# Biomass Smoke Exposure Reduces DNA Methylation Levels in *PRSS23* (cg23771366) in Women with Chronic Obstructive Pulmonary Disease

**DOI:** 10.3390/toxics13040253

**Published:** 2025-03-28

**Authors:** Gloria Pérez-Rubio, Ramcés Falfán-Valencia, Omar Andrés Bravo-Gutiérrez, Nancy Lozano-González, Alejandra Ramírez-Venegas, Filiberto Cruz-Vicente, María Elena Ramírez-Díaz

**Affiliations:** 1HLA Laboratory, Instituto Nacional de Enfermedades Respiratorias Ismael Cosío Villegas, Mexico City 14080, Mexico; rfalfanv@iner.gob.mx (R.F.-V.); a.bravo.gtz@gmail.com (O.A.B.-G.); nancylozano030@gmail.com (N.L.-G.); 2Tobacco Smoking and COPD Research Department, Instituto Nacional de Enfermedades Respiratorias Ismael Cosío Villegas, Mexico City 14080, Mexico; aleravas@hotmail.com; 3Internal Medicine Department, Hospital Civil Aurelio Valdivieso, Servicios de Salud de Oaxaca, Oaxaca 68050, Mexico; filitv6cv@hotmail.com; 4Coordinación de Vigilancia Epidemiológica, Jurisdicción 06 Sierra, Tlacolula de Matamoros Oaxaca, Servicios de Salud de Oaxaca, Oaxaca 70400, Mexico; drmariel2504@hotmail.com

**Keywords:** COPD, biomass-smoke, DNA methylation, PRSS23

## Abstract

COPD induced by biomass-burning smoke is a public health problem in developing countries. Biomass-based fuels are ineffective and deliver elevated levels of carbon monoxide, polycyclic aromatic hydrocarbons, and fine particulate matter. *PRSS23* participates in extracellular matrix remodeling processes in COPD patients. Our objective was to estimate the DNA methylation levels of cg23771366 (*PRSS23*) and their clinical relevance in COPD caused by chronic exposure to biomass-burning smoke (BBS). We included 80 women with COPD (COPD-BBS) (≥200 h per year), 180 women with exposure to BBS (≥200 h per year) but without COPD (BBES), and 79 lung-healthy women (HW) without exposure to biomass-burning smoke. The DNA methylation analysis shows significant differences between the three groups included in this study (*p* < 0.001). HW had high methylation levels (100%) in cg23771366 (*PRSS23*). In comparison, COPD-BBS and BBES had low levels [0.91% vs. 9.17%, respectively], showing statistically significant differences (*p* = 0.011) between both groups, with the COPD-BBS presenting the lowest levels in the methylation of cg23771366. In conclusion, chronic biomass-burning smoke exposure is associated with decreased levels of DNA methylation at the CpG cg23771366 site in *PRSS23*, reinforcing the relationship between *PRSS23* and particulate matter.

## 1. Introduction

Chronic Obstructive Pulmonary Disease (COPD) is a worldwide health problem. It is a complex and multifactorial disease; tobacco smoking (TS) is the leading risk factor for COPD. In addition, exposure to smoke produced from burning biomass contributes to COPD [1]. About 90% of COPD-related deaths occur in middle- and low-income countries, where there is high exposure to smoke from biomass burning. The principal population affected is women who use biomass fuels for cooking or heating their homes in poorly ventilated environments [2].

Biomass fuels are inefficient and produce high levels of pollutants such as carbon monoxide (CO), polycyclic aromatic hydrocarbons (PAHs), fine particulate matter with an aerodynamic diameter < 10 μm (PM10) or <2.5 μm (PM2.5), chlorides, sulfates, nitrates, aldehydes, benzene, nitrogen oxides, sulfur oxides, and free radicals [3]. PM2.5 can access the respiratory tract and be deposited in the airway and lung tissue. This leads to enhanced inflammatory response and impaired epithelial cell function [4]. Chronic exposure to PM2.5 causes impairment in lung function, emphysematous lesions, and remodeling of the airway walls [5,6]. In vitro assay in L02 cells (normal hepatocytes) shows that serine protease 23 (PRSS23) and zinc transporter ZIP10 (SLC39A10) might be potential biomarkers of PM2.5-induced carcinogenesis [7]. *PRSS23* is expressed in the nervous system, the lungs, the intestine, the pancreas, the skin, the kidneys, the heart, bone, and muscle. The expression of *PRSS23* has been associated with tumor progression in humans, the regulation of cellular proliferation, and cancer [8]. Through genome-wide association studies, PRSS23 has been found to participate in extracellular matrix remodeling processes in COPD patients [9,10].

A decrease in methylation at specific CpG sites in *PRSS23* (cg14391737, cg10711136, cg11660018, cg00475490, and cg23771366) has been associated with increased urinary total nicotine in tobacco smokers [11,12]. Earlier, we assessed the DNA methylation levels of cg23771366 in patients with a nicotine addiction; however, we did not observe significant differences [13].

There is significant evidence of differences between TS-COPD and BS-COPD in terms of genetic susceptibility, epigenetic factors, inflammation [1], clinical characteristics [14], tomographic patterns [15], and other factors. Thus, we aimed to assess the DNA methylation levels of cg23771366 (*PRSS23*) and their clinical significance in patients with BS-COPD.

## 2. Materials and Methods

We performed an observational and retrospective study at the Instituto Nacional de Enfermedades Respiratorias Ismael Cosío Villegas (INER) in Mexico City. We included 80 women with COPD (COPD-BBS) caused by chronic exposure to biomass-burning smoke (≥200 h per year), 180 women without COPD (BBES) but with an index of annual exposure to biomass-burning smoke of ≥200 h per year (BEI) [16], and 79 lung-healthy women (HW) without exposure to biomass-burning smoke. Current or former smoking women were excluded.

This protocol was approved by the INER research and biosafety bioethics committees (protocol code B03-23/28 February 2023). The individuals agreed to participate voluntarily and signed informed consent.

Participants completed a family history questionnaire, and biomass-burning smoke exposure was recorded to calculate the BEI. Peripheral blood samples were taken from each participant for subsequent DNA extraction [13].

The integrity of the DNA was confirmed following electrophoresis in a 1% agarose gel. The gel was inspected using a UV transilluminator, revealing bands. These solution-based genomic DNA purification kits ensure minimal DNA fragmentation, yielding DNA of up to 150 kb in size. The DNA presence and quality in each sample were verified with a NanoDrop 2000 spectrophotometer (Thermo Fisher Scientific, Waltham MA, USA). The quality control points included samples with at least ≥150 ng/μL and a 260/280 ratio between 1.8 and 2.0. Samples meeting the criteria were adjusted to 50 ng/μL for methylation evaluation.

The design of primers to assess cg23771366 in *PRSS23* was previously detailed [13]. The percentage of methylation was calculated using the EpiJET DNA Methylation Analysis Kit (Thermo Fisher Scientific, Waltham, MA, USA) [13]. The qPCR efficiencies were determined using LinRegPCR provided by the Real-Time PCR Data Markup Language (RDML) consortium [13,17].

The Kolmogorov–Smirnov test was used to assess normality for quantitative variables. We present the median, along with the 25th and 75th percentiles. Comparisons were performed using either the Mann–Whitney U-test or the Kruskal–Wallis test. We used Spearman’s correlation test to evaluate the relationship between methylation percentage and the biomass-burning smoke exposure index (BEI) or lung function test. A *p*-value of less than 0.05 was considered statistically significant in all our analyses.

## 3. Results

We included women exposed to biomass-burning smoke (≥200 h per year) with and without COPD and HW without exposure to biomass-burning smoke. HW were younger than women in the COPD-BBS and BBS groups; however, there was no statistically significant difference in age between the COPD-BBS and BBS groups (*p* = 0.081). The body mass index (BMI) does not show significant differences, nor does the biomass-burning smoke exposure index between COPD-BBS and BBES show significant differences.

Forced vital capacity (FVC), forced expiratory volume in the first second (FEV_1_), and the FEV_1_/FVC ratio show significant differences, with the COPD-BBS group having the lowest lung function values (Table 1). According to the Global Initiative for Chronic Obstructive Lung Disease (GOLD) guidelines 2025, in the COPD-BBS group, 30% of patients were classified as mild, 55% as moderate, and 15% as severe.

The DNA methylation analysis reveals significant differences among the three groups included in the study (*p* < 0.001). HW exhibited high levels of methylation (100%, 56.5–100.0), whereas COPD-BBS and BBES showed low levels of methylation at the CpG cg23771366 located in the PRSS23 gene [0.91 (0–18.97) vs. 9.17 (0.04–99.9), respectively], indicating statistically significant differences (*p* = 0.011) between the two groups, with COPD-BBS demonstrating the lowest levels of methylation at cg23771366 (Figure 1). The study population showed a statistical difference in age, and then using this variable as a potential confounding factor, the age-corrected *p*-value was *p* = 0.034.

We evaluated the correlation between BEI, pulmonary function tests, and the methylation percentage of the cg23771366 (*PRSS23*) site; however, no significant correlations were found (Table 2). Additionally, we did not observe significant differences in the methylation percentage among the GOLD stages in the patient group.

## 4. Discussion

We included women exposed to biomass-burning smoke, both with and without COPD, and women without respiratory issues who were not exposed. The COPD-BBS and BBS groups showed no significant differences in age, BMI, or BEI. Lung function was lower in the COPD-BBS group compared to the BEES and HW groups. Exposure to biomass-burning smoke from cooking or heating is a common feature among women in developing nations and rural areas. Smoke from biomass burning contains elevated levels of particulate matter that settle inside homes [14]. In these environments, up to 1000 µg/m^3^ of PM2.5 can enter the respiratory tract [18].

Long-term exposure to elevated concentrations of PM increases the prevalence of COPD and lung cancer in adults, resulting in a decline in pulmonary function [19]. An *in vitro* model has demonstrated that *PRSS23* expression positively correlates with the presence of CD8+ T cells, CD4+ cells, macrophages, neutrophils, and dendritic cells, suggesting a potential role for PRSS23 in regulating immune cells and metastasis [20]. High-throughput RNA sequencing has demonstrated that PRSS23 positively regulates the expression of fibroblast growth factor 2/fibroblast growth factor-binding protein 1 (FGF2/FGFBP1) in fibroblasts. This finding indicated that PRSS23 and FGF2 were significantly overexpressed in M2 macrophages [21]. The current study revealed differences in methylation levels at the CpG site cg23771366.

PRSS23 has been reported to be overexpressed in malignant pleural mesothelioma [22], in breast cancer cells that are positive for the ligand-binding domain of estrogen receptor-α [8], and in gastric cancer patients [21,23]; however, in Ewing sarcoma, the expression was reduced [24]. It is suggested that the expression levels of *PRSS23* be evaluated to determine how its methylation influences them. Additionally, it would be prudent to monitor the amount of particulate matter to which these women are exposed, which would corroborate whether PRSS23 could serve as a biomarker for such exposure [7]. Epigenetic modifications, such as DNA methylation, play a crucial role in determining the activation or silencing of genes. Chronic environmental exposures can alter the epigenetic profile of cells and contribute to gene transcription [25,26] (Figure 2).

It should be noted that studies on breast cancer cells indicated that *PRSS23* is upregulated by estrogen receptor 1 [1]. Reports have shown that among smokers, a distinct pattern of gene expression exists based on sex. In women, pathways related to immune and inflammatory responses are significant. Additionally, those associated with female hormones play a crucial role, notably PRSS23, which is linked to the beta-estradiol/estrogen pathway [27]. *PRSS23* has been associated with various cancers, with high expression levels typically correlating with a worse prognosis. Hsa-miR-7-5p has been identified as a regulator of this gene and *SERPINA1* [28]. The *SERPINA1* gene encodes the alpha-1 antitrypsin protein, which primarily protects the lower respiratory tract of the lungs from proteolytic degradation by neutrophil elastase [29]. PRSS23’s role in the lungs has been minimally explored; however, considering this context, it is worthwhile to investigate its function further, particularly in relation to hormonal influences.

It is worth noting that among the study’s main limitations are other environmental factors that can modify methylation levels, such as diet and lifestyle; however, the study population did not have statistically significant differences in body mass index and bio-mass smoke exposure index; also, cases and controls belong to the same region, suggesting that their diet and lifestyle are similar. Our study employed a cross-sectional design, which precluded the determination of a causal relationship between biomass smoke exposure and DNA methylation levels. Future studies could consider longitudinal designs to better understand the temporal relationship between exposure and changes in methylation. We suggest that gene expression should be evaluated in the lung, in addition to assessing gene methylation.

## 5. Conclusions

Exposure to biomass-burning smoke is associated with reduced DNA methylation levels at the CpG site cg23771366 in *PRSS23*, further highlighting the connection between PRSS23 and particulate matter. Patients with COPD exhibited the lowest percentages of methylation.

## Figures and Tables

**Figure 1 toxics-13-00253-f001:**
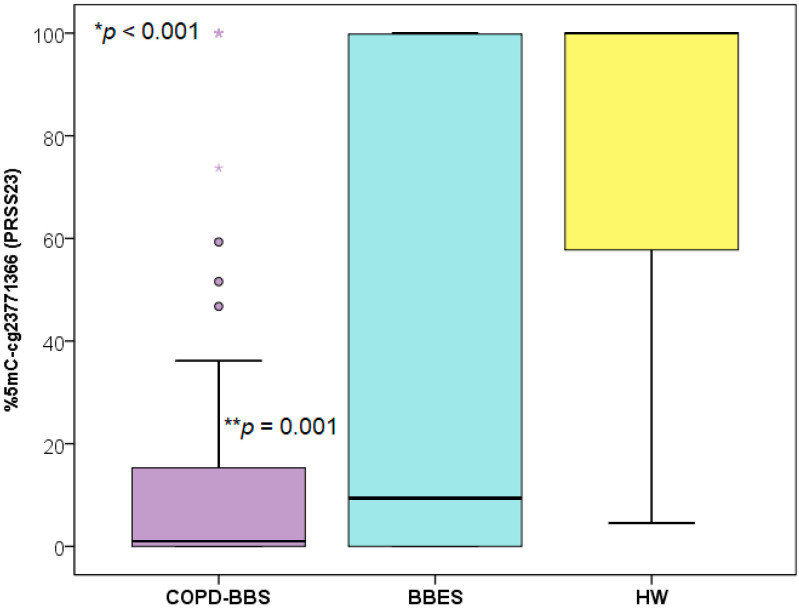
Methylation percentage between COPD-BBS, BBES, and HW at CpG cg23771366 (*PRSS23*) site. * The *p*-value by Kruskal–Wallis test. The *p*-value was determined using the ** Mann–Whitney U test comparing COPD-BBS with BBES. The *p*-value = 0.034 corrected by age.

**Figure 2 toxics-13-00253-f002:**
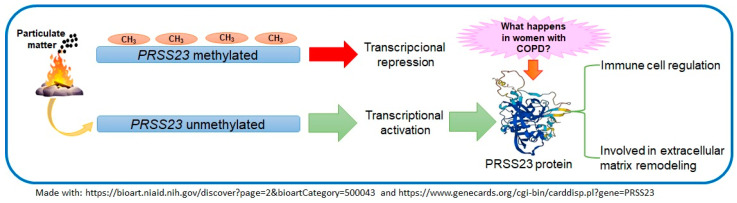
Proposal on the involvement of *PRSS23* and methylation levels in COPD.

**Table 1 toxics-13-00253-t001:** Demographic and clinical characteristics of participants.

Variable	COPD-BBS (*n* = 80)	BBES (*n* = 180)	HW (*n* = 79)	*p*-Value
Age (years)	66 (58–74) *	61 (56–69)	50 (44–60)	<0.001 *
BMI (kg/m^2^)	26.5 (24.1–30.4)	28.4 (25.0–31.7)	27.4 (24.6–28.9)	0.080 *
BEI (h/year)	300 (205–365)	270 (200–365)	NA	0.306 **
FVC (%)	87 (77–105)	97 (88–110)	88 (77–96)	0.009 *
FEV_1_ (%)	70 (59–83)	104 (95–116)	100 (82–106)	<0.001 *
FEV_1_/FVC	61 (54–67)	84 (80–88)	86 (79–91)	<0.001 *

COPD-BBS: women with COPD secondary to chronic exposure to biomass-burning smoke. BBES: women exposed to biomass-burning smoke but without COPD. HW: women with healthy lungs without exposure to biomass-burning smoke. BMI: body mass index. BEI; biomass-burning smoke exposure index. FVC: forced vital capacity. FEV_1_: forced expiratory volume in the first second. NA; Not applicable. * *p*-value by Kruskal–Wallis test or ** Mann–Whitney U-test. Median and percentiles 25 and 75 are shown.

**Table 2 toxics-13-00253-t002:** Spearman’s correlation between levels of methylation at cg23771366 (*PRSS23*) and variables like exposure to biomass-burning smoke and lung function in women with COPD.

Variable	BEI (h/Year)	FVC %	FEV1 %	FEV1/FVC
Spearman’s rho	0.015	0.096	−0.077	−0.150
*p*-value	0.897	0.408	0.504	0.183

BEI: biomass-burning smoke exposure index. FVC: forced vital capacity. FEV_1_: forced expiratory volume in the first second.

## Data Availability

The original contributions presented in the study are included in the article. Further inquiries can be directed at the corresponding author.

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
