# Peer review of "Biomass Smoke Exposure Reduces DNA Methylation Levels in PRSS23 (cg23771366) in Women with Chronic Obstructive Pulmonary Disease"

_toxics, 2025, doi:10.3390/toxics13040253_

Round 1
Reviewer 1 Report
Comments and Suggestions for Authors
Title: Biomass smoke exposure reduces DNA methylation levels in 2 PRSS23 (cg23771366) in women with COPD The manuscript explores the relationship between biomass combustion smoke exposure and chronic obstructive pulmonary disease (COPD), which is an important public health issue, especially in developing countries. The study focused on the female group who face higher health risks from long-term use of biomass fuels for cooking or heating. The study included 80 patients with COPD (COPD-BBS), 180 women who were exposed to biomass smoke but did not develop COPD (BBES), and 79 healthy women who were not exposed to biomass smoke (HW). The study has a relatively large sample size and reasonable grouping. The study found that COPD patients (COPD-BBS) exposed to biomass smoke over a long period had significantly reduced DNA methylation levels at cg23771366 of the PRSS23 gene, with the lowest methylation levels in the COPD-BBS group. I recommend the authors should address the following issues.
- Although methylation levels of PRSS23 were found to be associated with biomass smoke exposure, the study did not delve into the specific molecular mechanisms. For example, how methylation of PRSS23 affects its expression, and how this expression change contributes to the pathological process of COPD.
- The study focused mainly on biomass smoke exposure but did not take into account other environmental factors that may affect DNA methylation, such as air pollutants including NO2, O3, diet, lifestyle, etc. This confounder may interfere with the findings.
- The statistical methods were not verified by multiple tests. The limitations of the statistical methods may lead to the risk of false positive results.
- The study was a cross-sectional design and could not determine a causal relationship between biomass smoke exposure and DNA methylation levels. Future studies could consider longitudinal designs to better understand the temporal relationship between exposure and methylation changes.
- The study should discuss the limitations of the study design including sample selection bias, and potential confounding factors.
Author Response
Dear reviewer.
We appreciate your comments and respond to each of them below.
Reviewer 1
Title: Biomass smoke exposure reduces DNA methylation levels in 2 PRSS23 (cg23771366) in women with COPD The manuscript explores the relationship between biomass combustion smoke exposure and chronic obstructive pulmonary disease (COPD), which is an important public health issue, especially in developing countries. The study focused on the female group who face higher health risks from long-term use of biomass fuels for cooking or heating. The study included 80 patients with COPD (COPD-BBS), 180 women who were exposed to biomass smoke but did not develop COPD (BBES), and 79 healthy women who were not exposed to biomass smoke (HW). The study has a relatively large sample size and reasonable grouping. The study found that COPD patients (COPD-BBS) exposed to biomass smoke over a long period had significantly reduced DNA methylation levels at cg23771366 of the PRSS23 gene, with the lowest methylation levels in the COPD-BBS group. I recommend the authors should address the following issues.
- Although methylation levels of PRSS23 were found to be associated with biomass smoke exposure, the study did not delve into the specific molecular mechanisms. For example, how methylation of PRSS23 affects its expression, and how this expression change contributes to the pathological process of COPD.
Thank you for your comment. We added Figure 2 (lines 168-170) to explain the methylation mechanism on transcription and its probable role in COPD. We also added a reference that describes the relevance of epigenetics in gene regulation (lines 164-168)
- The study focused mainly on biomass smoke exposure but did not take into account other environmental factors that may affect DNA methylation, such as air pollutants including NO2, O3, diet, lifestyle, etc. This confounder may interfere with the findings.
Thank you for the comment. We've added references to the severity of indoor pollution and its clear association with COPD risk.
You are correct; diet and lifestyle can affect methylation levels. In the present report, BMI between the study groups did not show statistically significant differences, and both groups were recruited from rural or suburban areas (lines 147-148). However, they are described in the limitations section to be considered (lines 184-188)
- The statistical methods were not verified by multiple tests. The limitations of the statistical methods may lead to the risk of false positive results.
Thank you for the observation. We corrected the p-value by age because it is a variable with significative differences between COPD-BBS and BBES; now, the p= 0.034 (lines 124-126)
- The study was a cross-sectional design and could not determine a causal relationship between biomass smoke exposure and DNA methylation levels. Future studies could consider longitudinal designs to better understand the temporal relationship between exposure and methylation changes.
Thank you for the comment. This limitation was added (lines 184-193)
- The study should discuss the limitations of the study design including sample selection bias, and potential confounding factors.
Thank you, we added the section limitations (lines 193-200).

Reviewer 2 Report
Comments and Suggestions for Authors
In this short report, Pérez-Rubio and colleagues describe the presence of hypomethylation of a specific CpG site of the PRSS23 gene in peripheral blood cells of women exposed to biomass burning smoke. Such smoke exposure was due to biomass-based cooking or heating activities, which are common in less developed areas. Women with COPD had the lowest levels of 5mC-methylation despite being exposed to the same amount of smoke per year as women without COPD, whereas healthy women not exposed to smoke had high levels of PRSS23 gene methylation.
Main
Two groups of women included in this study were selected on the basis of chronic exposure to more than 200 hours of biomass burning smoke per year. However, it is unclear for how many years they were exposed to such high levels of smoke. Could an index similar to the "pack year" used for cigarette smoke exposure be developed and used to assess possible correlations with the measured parameters?
In addition, given the possibility of interaction, were active tobacco smokers excluded from the study or otherwise considered in subgroup analyses?
Was the expression of the PRSS23 gene in the blood cells of the patients studied assessed and related to its specific CpG methylation? There is a brief mention in the Discussion section (lines 157-158) that the expression levels of PRSS23 should be assessed. If this was not done, please mention this as a limitation of the study.
What correlation analysis tests were performed to evaluate the correlations between cg23771366-PRSS23 methylation percentage and lung function variables? Please describe them in the Methods section and consider including a table showing the results obtained and their p-values, even if they are not statistically relevant.
The Discussion section could be revised to focus more on the results obtained. In particular, the first paragraph seems to be descriptive and does not give a clear message to the reader. The limitations of the study could also be acknowledged.
Minor
Please provide an appropriate reference describing the experimental conditions used to perform qPCR to assess the expression of the cg23771366-PRSS23 gene, as this appears to be missing from line 95 of the text.
Please add the methylation results of the HW group to Figure 1 and the associated statistical significance.
Please check the results shown in Figure 1 as they appear slightly different from those reported in the main text (line 127). In particular, the 75th percentile values for both groups appear different in the main text than in the figure.
Author Response
Dear review.
We appreciate your comments and respond to each of them below.
Reviewer 2
In this short report, Pérez-Rubio and colleagues describe the presence of hypomethylation of a
specific CpG site of the PRSS23 gene in peripheral blood cells of women exposed to biomass burning
smoke. Such smoke exposure was due to biomass-based cooking or heating activities, which are
common in less developed areas. Women with COPD had the lowest levels of 5mC-methylation
despite being exposed to the same amount of smoke per year as women without COPD, whereas
healthy women not exposed to smoke had high levels of PRSS23 gene methylation.
Main
1. Two groups of women included in this study were selected on the basis of chronic exposure
to more than 200 hours of biomass-burning smoke per year. However, it is unclear for how
many years they were exposed to such high levels of smoke. Could an index similar to the
"pack year" used for cigarette smoke exposure be developed and used to assess possible
correlations with the measured parameters?
We add the reference about the index of annual exposure to biomass-burning smoke (≥ 200
hours per year). This paper shows that exposure of 200-hour-year to biomass smoke has an
OR of 15 for the risk of COPD (lines 70-72).
2. In addition, given the possibility of interaction, were active tobacco smokers excluded from
the study or otherwise considered in subgroup analyses?
Lines 70-73. We added, “Current or former smokers women were excluded.”
3. Was the expression of the PRSS23 gene in the blood cells of the patients studied assessed
and related to its specific CpG methylation? There is a brief mention in the Discussion
section (lines 157-158) that the expression levels of PRSS23 should be assessed. If this was
not done, please mention this as a limitation of the study.
We added this observation (lines 188-190).
4. What correlation analysis tests were performed to evaluate the correlations between
cg23771366-PRSS23 methylation percentage and lung function variables? Please describe
them in the Methods section and consider including a table showing the results obtained
and their p-values, even if they are not statistically relevant.
Lines 95-97. We described the correlation analysis; we added table 2 with the results
obtained (lines 135-139)
5. The Discussion section could be revised to focus more on the results obtained. In particular,
the first paragraph seems to be descriptive and does not give a clear message to the reader.
The limitations of the study could also be acknowledged.
We corrected the first paragraph (lines 144-148) and added limitations section (lines 184-
193)
Minor
1. Please provide an appropriate reference describing the experimental conditions used to
perform qPCR to assess the expression of the cg23771366-PRSS23 gene, as this appears to
be missing from line 95 of the text.
Thank you, we added the reference 13
2. Please add the methylation results of the HW group to Figure 1 and the associated statistical
significance.
Thank you. We add the HW group to Figure 1
3. Please check the results shown in Figure 1 as they appear slightly different from those
reported in the main text (line 127). In particular, the 75th percentile values for both groups
appear different in the main text than in the figure.
Thank you. We checked the results and corrected Figure 1

Round 2
Reviewer 1 Report
Comments and Suggestions for Authors
accept